# Simulation-based inference for non-parametric statistical comparison of biomolecule dynamics

**Hippolyte Verdier**[1,2,3]*, **François Laurent**[1], **Alhassan Cassé**[3], **Christian L. Vestergaard**[1], **Christian G. Specht**[4]*, **Jean-Baptiste Masson**[1]*

**1** Institut Pasteur, Université Paris Cité, CNRS UMR 3751, Decision and Bayesian Computation, Paris, France, **2** Université de Paris, UFR de physique, Paris, France, **3** Histopathology and Bio-Imaging Group, Sanofi, R&D, Vitry-Sur-Seine, France, **4** Diseases and Hormones of the Nervous System (DHNS), Inserm U1195, Université Paris-Saclay, Paris, France

* hverdier@pasteur.fr (HV); christian.specht@inserm.fr (CS); jbmasson@pasteur.fr (J-BM)

**Data Availability Statement:** 1) GRATIN encoder: https://github.com/hippover/gratin 2) Palmari (image processing, from.tif files to trajectories): https://github.com/hippover/palmari/ 3) MMD

## Abstract

Numerous models have been developed to account for the complex properties of the random walks of biomolecules. However, when analysing experimental data, conditions are rarely met to ensure model identification. The dynamics may simultaneously be influenced by spatial and temporal heterogeneities of the environment, out-of-equilibrium fluxes and conformal changes of the tracked molecules. Recorded trajectories are often too short to reliably discern such multi-scale dynamics, which precludes unambiguous assessment of the type of random walk and its parameters. Furthermore, the motion of biomolecules may not be well described by a single, canonical random walk model. Here, we develop a two-step statistical testing scheme for comparing biomolecule dynamics observed in different experimental conditions without having to identify or make strong prior assumptions about the model generating the recorded random walks. We first train a graph neural network to perform simulation-based inference and thus learn a rich summary statistics vector describing individual trajectories. We then compare trajectories obtained in different biological conditions using a non-parametric maximum mean discrepancy (MMD) statistical test on their so-obtained summary statistics. This procedure allows us to characterise sets of random walks regardless of their generating models, without resorting to model-specific physical quantities or estimators. We first validate the relevance of our approach on numerically simulated trajectories. This demonstrates both the statistical power of the MMD test and the descriptive power of the learnt summary statistics compared to estimates of physical quantities. We then illustrate the ability of our framework to detect changes in $\alpha$-synuclein dynamics at synapses in cultured cortical neurons, in response to membrane depolarisation, and show that detected differences are largely driven by increased protein mobility in the depolarised state, in agreement with previous findings. The method provides a means of interpreting the differences it detects in terms of single trajectory characteristics. Finally, we emphasise the interest of performing various comparisons to probe the heterogeneity of experimentally acquired datasets at different levels of granularity (e.g., biological replicates, fields of view, and organelles).

comparisons: We made the code of the palm-tools repo accessible to the public: https://gitlab.pasteur.fr/hverdier/palm-tools 4) We also released a web platform (currently in beta) allowing researchers to perform the kind of analysis presented in this paper on their data: https://tracktor.pasteur.cloud/ 5) Notebook where the analysis is performed, using the previously mentioned codes: https://gitlab.pasteur.fr/hverdier/palm-tools/-/blob/master/examples/MMD-paper.ipynb.

**Funding:** This study was funded by the Institut Pasteur, L'Agence Nationale de la Recherche (TRamWAy, ANR-17-CE23-0016 to JBM), the INCEPTION project (PIA/ANR-16-CONV-0005, OG), and the "Investissements d'avenir" programme under the management of Agence Nationale de la Recherche, reference ANR-19-P3IA-0001 (PRAIRIE 3IA Institute) to JBM & CLV. The funding sources had no role in study design, data collection and analysis, 613 decision to publish, or preparation of the manuscript.

**Competing interests:** I have read the journal's policy and the authors of this manuscript have the following competing interests: HV and AC are Sanofi employees and may hold shares and/or stock options in the company. The other authors declare to have no financial or non-financial conflicts of interest.

## Author summary

The continuous improvement of methods for single molecule tracking in live cells are driving our understanding of how biomolecules move inside cells. Analysing trajectories of single molecules is complicated by their highly erratic and noisy nature and thus requires the use of statistical models of their motion. However, it is often not possible to unambiguously determine a model from a set of short and noisy trajectories. Furthermore, the heterogeneous nature of the cellular environment means that the molecules' motion is often not properly described by a single model. In this paper we develop a new statistical testing scheme to detect differences in biomolecule dynamics within organelles without needing to identify a model of their motion. We train a graph neural network on large-scale simulations of random walks to learn a latent representation that captures relevant physical properties of a trajectory. We use a kernel-based statistical test within that latent space to compare the properties of two sets of trajectories recorded under different biological conditions. We apply our approach to detect differences in the dynamics of $\alpha$-synuclein, a presynaptic protein, in axons and boutons during synaptic stimulation. This represents an important step towards automated single-molecule-based read-out of pharmacological action.

This is a *PLOS Computational Biology* Methods paper.

## Introduction

The analysis of biomolecule trajectories in general focuses either on the estimation of predefined descriptive statistics or on inference of an assumed generative model's parameters. These quantitative descriptions are often subsequently used to compare trajectories observed in different biological conditions or at different locations in the cell. The heterogeneity typically exhibited within experimental observations of biomolecule dynamics, as well as technological constraints limiting the amount of accessible information (e.g. limiting trajectory lengths, spatial tracer densities, and localisation precision) has motivated the development of specific analysis tools for trajectories observed in biological samples using single molecule localization microscopy (for a review, see [1]). Indeed, while other fields of physics, such as condensed matter, have a rich literature on random walks [2, 3], their methods are not directly applicable to the analysis of experimentally observed biomolecule trajectories. These often cannot be assumed to be realisations of neither stationary nor ergodic stochastic processes and are in general several orders of magnitude shorter than those observed in physical systems or obtained via simulations.

Biomolecule trajectories are most frequently described by their mean square displacement curve, from which the diffusion coefficient and the anomalous diffusion exponent can notably be estimated (see [4, 5] for reviews of these techniques). However, these method suffer from several flaws as discussed in [1]. Therefore, many tailored inference schemes focusing on the estimation of parameters of an assumed generative model have also been proposed, e.g. the transition rates between modes of free diffusion and the associated diffusion coefficients [6, 7] or the forces and diffusion coefficients of Langevin dynamics [8, 9]. Besides, previous work have shown how carefully chosen features could allow one to distinguish between two modes

of diffusion: [10], for instance, proposes a test to distinguish continuous time random walks (CTRW) from fractional Brownian motion (fBM) based on a metric derived from the generalization of the total variation, while [11] demonstrates how the scaling law of the mean maximal excursion allows one to differentiate between trajectories of CTRW, fBM and diffusion on fractals. Similar approaches have been proposed to detect change-points in intermittent stochastic processes, based for instance on the random walks' convex hull [12].

Nevertheless, no general statistical framework allows, to our knowledge, to assess the statistical significance of observed differences between sets of biomolecule trajectories without first assuming a given generative model or restricting to a certain set of descriptive statistics, often chosen according to an *a priori* knowledge of the system. The method we propose here aims to fill this gap by allowing to detect differences between experimentally recorded sets of biomolecule trajectories, assess their relative magnitudes and statistical significance, without the need for prior information on the expected type of dynamics. It is therefore expected to be especially useful in cases where little is known about the observed biological system.

The method can be split in two independent parts: the first consists in obtaining a good descriptive statistics of the trajectories, while the second consists in detecting differences between the values of these statistics and assessing their statistical significance. In this work, we obtain descriptive statistics via simulation-based inference [13] and use the non-parametric maximum mean discrepancy (MMD) test to assess the statistical significance of the differences [14]. However, it is to be noted that the method is adaptable: if one is interested in a precise aspect of the dynamics, tailored descriptions of trajectories can be used instead of the generalist latent description proposed here. Similarly, one may also use other statistical tests to test for specific types of variability. It is important to remark that the process through which we obtain descriptive statistics of the trajectories is not explicitly optimized for the subsequent statistical test. The independence of these two steps is required to control the type I error of the test.

## Descriptive vectors obtained via simulation-based inference

Numerical models allow us to generate synthetic observations of biological systems across a broad range of parameters. However, the computational cost of directly using these simulations to perform statistical inference is often prohibitive [13]. The reason is that both the likelihood and evidence are intractable in most systems, and these inferences must therefore be addressed as likelihood-free simulation-based ones [15]. The primary approach to simulation-based inference is approximate Bayesian computation (ABC), which relies on comparing user-defined summary statistics from experimentally recorded and simulated data using a chosen distance metric. The ever-growing amount of available data, along with recent advances in deep learning [16] make it possible to capture more and more detailed properties of experimental systems, and have thus boosted the development of simulation-based inference. We refer the interested reader to [13] for a detailed taxonomy of simulation-based inference methods.

Recent works, notably motivated by the AnDi challenge [17], have shown that deep-learning methods yield more precise estimates of random walk parameters than analytic ones [18–20]: estimators trained on simulated trajectories are more versatile, robust to various sources of noise and to biases stemming from the limited length of experimental recordings (often shorter than 15 localizations), which hamper the estimation of asymptotically-defined quantities. To achieve this performance, deep networks learn a rich *latent* description of random walk trajectories. Hence, instead of describing trajectories using estimates (analytic or resorting to neural networks) of these explicitly defined features (be it model parameters or other statistics), we instead chose to rely directly on the *latent vector* learnt by the *encoder* module of

a neural network trained to estimate physical variables. This choice is motivated both by the diversity of the dynamics that it allows one to distinguish and by the recently demonstrated superiority of deep-learning methods over analytical ones to estimate properties of short random walks observed with experimental noise. A recent study took a slightly different approach to characterizing random walks without relying on explicit descriptive statistics, using auto-encoders trained to reconstruct the trajectories of different random walk models and treating the analysis of experimental random walks as an anomaly detection problem [21].

### Non-parametric test to compare descriptive statistics

Here, to compare trajectories using the aforementioned description vectors, we use the maximum mean discrepancy (MMD) introduced in [14] and its associated permutation test. This non-parametric and kernel-based statistical test allows detecting eventual differences of dynamics between pairs of sets of trajectories. We show on simulated trajectories that such tests, based on the learnt description vectors, are more sensible than those based on estimates of commonly used variables, and than $t$–tests based on these estimates, classically used to assess the significance of such differences. This methodology can in particular be used to compare dynamics observed in different biological conditions and cell organelles. Its central advantage is that it is not dependent on the prior specification and selection of a model of the recorded random walks—a choice which would restrict the scope of comparisons by focusing only on certain parameters. This enables generic and statistically robust comparison of differing biological conditions, which are likely to induce different levels of cellular heterogeneity and do not necessarily generate canonical random walks.

### Application to the dynamics of $\alpha$-synuclein

We validate our methodology by studying the dynamics of $\alpha$-synuclein inside and outside of synapses. $\alpha$-synuclein is a small, soluble, and highly mobile protein (140 amino acid residues) that is strongly accumulated in presynaptic boutons (reviewed in [22]). Experiments based on fluorescence recovery after photobleaching [23] have shown the existence of at least two main modes of diffusion, one in which $\alpha$-synuclein is transiently bound to synaptic vesicles in the synaptic bouton, and another in which the protein diffuses freely both in axons and in synaptic regions. The existence of an immobile population of $\alpha$-synuclein molecules, taking the form of protein aggregates at synapses, has also been proposed [23]. In response to strong depolarising signals the bound population of $\alpha$-synuclein dissociates from its synaptic binding sites and disperses in the neighbouring axon [24]. In agreement with these earlier studies, we found that $\alpha$-synuclein dynamics differ between synapses and axons. Furthermore, depolarisation of the neurons shifts the relative frequency of the proteins from a less mobile to a highly mobile state, but it does not appear to induce qualitative changes in the type of diffusion dynamics the molecules follow. It is not yet clear what role this dynamic shift of $\alpha$-synuclein plays in vesicle cycling and in the regulation of synaptic transmission. Single molecule based imaging in living neurons can help to address this question and yield new information about the physiological function of $\alpha$-synuclein at synapses, as well as its involvement in pathological processes.

## Materials and methods

### Recording $\alpha$Syn:Eos4 dynamics

**Neuron cultures and $\alpha$Syn:Eos4 expression.**   Primary murine cortical neuron cultures were prepared at embryonic day E17 as described previously [25]. Cortices were dissected, the tissue was dissociated and the cells were seeded at a concentration of $5 \times 10^4 \, \mathrm{cm}^{-2}$ on glass

coverslips that had been coated with poly-D,L-ornithine. Neurons were kept at 37˚C and 5% $CO_2$ in neurobasal medium supplemented with Glutamax, antibiotics and B27 (all from Gibco, Thermo Fisher Scientific), infected at day *in vitro* (DIV) 11–24 with lentivirus driving the expression of $\alpha$-synuclein tagged at its C-terminus with the photoconvertible fluorescent protein mEos4b ($\alpha$Syn:Eos4) under the control of a ubiquitin promotor, and used for experiments 7 days later. All cell culture and imaging experiments were conducted at the Laboratory for cellular synapse biology at IBENS (Paris). Procedures involving animals were performed according to the guidelines set out by the local veterinary and administrative authorities.

**Single molecule localisation microscopy (SMLM).**   Living neurons expressing $\alpha$Syn:Eos4 were imaged in modified Tyrode's solution (in [mM]: 120 NaCl, 2.5 KCl, 2 CaCl$_2$, 2 MgCl$_2$, 25 glucose, 5 pyruvate, 25 HEPES, adjusted to pH 7.4) at room temperature, using an inverted Nikon Eclipse Ti microscope equipped with a 100x oil objective (NA 1.49), an Andor iXon EMCCD camera (16 bit, image pixel size 160 nm), and Nikon NIS acquisition software. First, an image of the chosen field of view (average of 10 image frames taken with 100 ms exposure time) was taken in the green channel (non-converted mEos4b fluorescence) using a mercury lamp and specific excitation (485/20 nm) and emission filters (525/30 nm). This was followed by a streamed acquisition of 25 000 movie frames recorded with 15 ms exposure and $\Delta t$ = 15.4 ms time lapse (total duration: 6 min 25 s) in the red channel using a 561 nm laser at a nominal power of 150 mW for excitation (inclined illumination), together with pulsed activation lasers applied during the off time of the camera (405 nm, approx. 1–5 mW; 488 nm, 10 mW; 0.45 ms pulse). The red emission of the photo-converted mEos4b fluorophores was detected with a 607/36 nm filter (Fig 1A).

After recording of the baseline dynamics of $\alpha$Syn:Eos4, the buffer composition was changed with the addition of Tyrode's solution containing elevated KCl at the expense of NaCl (final concentrations in [mM]: 78 NaCl, 44.5 KCl, 2 CaCl$_2$, 2 MgCl$_2$, 25 glucose, 5 pyruvate, 25 HEPES, pH 7.4). This treatment causes the depolarisation of the neurons leading to the dissociation of $\alpha$-synuclein from its binding sites in the synaptic bouton [24]. A reference image was taken in the green channel, followed by a second SMLM recording (Fig 1B), starting approximately 7.5 min after the first acquisition. Finally, the neurons were fixed with the addition of phosphate buffer at pH 7.4 containing 4% paraformaldehyde and 1% sucrose (final concentration 2% PFA), and a third reference image (green) and SMLM movie (red channel) were acquired in the presence of the fixative (Fig 1C).

**Image processing and analysis.**   SMLM image stacks (tiff files) were pre-processed in order to remove background fluorescence using a quantile filter computed on a sliding window. Then, localisations were detected using the algorithm described in [26], based on a wavelet analysis. Subpixel localisation was performed using the radial symmetry center algorithm introduced in [27]. Sample drift was corrected by subtracting the displacement that yielded the best correlation between densities of successive temporal slices grouping 10 000 localisations each. To isolate axons, we applied a Sato filter [28] with a width of 3 pixels on the logarithm of the pixel-wise mean intensity. Then, we used a local thresholding algorithm, provided by [29] to compute a mask over the image. All steps of the processing were implemented in Python using the `palmari` package. Synapses were manually detoured using an ad-hoc graphic user interface. In total, our analysis includes 321 synapses for which more than 150 trajectories were recorded, coming from 10 different fields of view. A synapse appearing in several recordings based on the same field of view but done in different conditions (e.g. some synapses appear in control, KCl and fixed conditions) is counted several times.

In the analysis of experimental trajectories, we considered only trajectories located in the axons, and we split them into two groups: those located outside the synaptic region and those located inside. Synaptic regions were delimited by a density threshold of one tenth of the

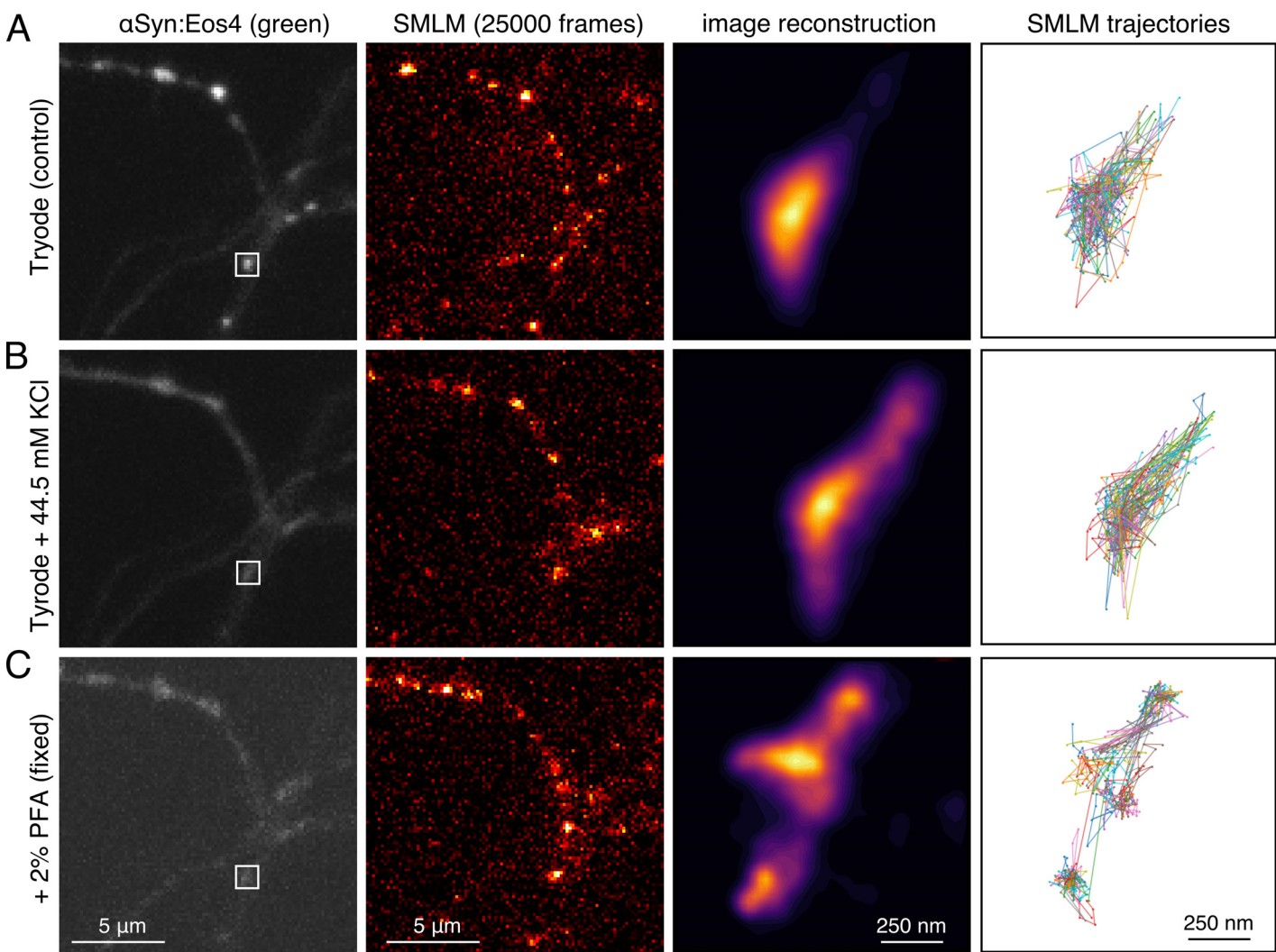

**Fig 1. Single molecule localisation microscopy (SMLM) of $\alpha$-synuclein in cortical neurons.** (**A**) Neurons expressing $\alpha$Syn:Eos4 were first imaged in control condition. A reference image (left panel) was taken in the green channel, followed by a SMLM movie of 25000 frames in the red channel (panel two). (**B**) A second recording of the same field of view (image and movie) were then acquired in the presence of elevated KCl concentration. Note the dispersal of $\alpha$Syn:Eos4 in response to depolarisation compared to the control (left panels). (**C**) A third image and movie were acquired after addition of 2% paraformaldehyde. The third column of images shows zoomed SMLM reconstructions of the synaptic bouton indicated in the first image. The fourth column depicts a subset of trajectories from the same synaptic terminal.

maximum density of detections in the synapse. The density was estimated using a Gaussian kernel method with a bandwidth of 150 nm. We estimated the apparent effective diffusivity [30] of a trajectory from the sample variance of its single-time-lapse displacements, i.e. $\hat{D} = \sum_{i=1}^{N} ||\Delta\mathbf{r}_i - \boldsymbol{\mu}||^2)/[4(N-1)\Delta t]$, where $\Delta\mathbf{r}_i = \mathbf{r}_i - \mathbf{r}_{i-1}$ is the displacement between the $(i-1)$th and $i$th recorded positions and $\boldsymbol{\mu} = \sum_{i=1}^{N} \Delta\mathrm{r}_i/N$ is the average displacement.

## Trajectory descriptions with latent vectors

We propose here a new method to obtain descriptive statistics of random walks, without requiring assumptions about the underlying generative models. In this section, we present the simulation-based inference scheme on which this method relies, the architecture of the neural

network used to compute latent vectors from trajectories, and the visualisation of these vectors.

**Simulation-based inference.**   In order to ensure that our characterisation of trajectories is accurate, robust and length-independent, it should be trained on an as wide as possible variety of random walks. Hence, we chose to rely on a simulation-based inference procedure [13], which consists in generating data on which the neural network is subsequently trained. In our case, this amounts to simulating trajectories of a variety of models known to exhibit a diverse set of properties of biomolecule dynamics in cells. The physical parameters chosen to simulate these trajectories should at least cover the range of the experimentally observed ones, in order to ensure that the network is able to encode relevant information about the recorded trajectories on which the inference will eventually be performed after its training.

To ensure the diversity of the training set, we simulated trajectories using five different canonical random walk models covering a wide spectrum of possible random walk characteristics:

- the Levy walk (LW) [31–33], which has non-Gaussian increments and exhibits weak ergodicity breaking;

- scaled Brownian motion (sBM) [34–36], which is Gaussian, non-stationary and weakly non-ergodic;

- the Ornstein Uhlenbeck process (OU) [37], a Gaussian, stationary process with exponentially decaying autocorrelations;

- fractional Brownian motion (fBM) [38], which is Gaussian, stationary and exhibits slowly decaying temporal correlations;

- and the continuous time random walk (CTRW) [10, 39], which is non-Gaussian, shows weak ergodicity breaking, ageing, and has discontinuous paths.

The models' parameters were drawn from the same distributions throughout the entire study and were chosen to cover the entire ranges observed experimentally. Trajectory lengths were drawn from a log-uniform distribution between 7 and 25 points, which corresponds to a mean length of 14 points. The effective diffusivity was drawn from a log-normal distribution with $D_0 = 1\mu m^2/s$, $\langle \log_{10}(D/D_0) \rangle = -0.5$ and $\mathrm{Var}(\log_{10}(D/D_0)) = 0.5^2$. For the OU model, the relaxation rate $\theta$ was log-uniformly drawn between 0.01 and 1.

We added uncorrelated localisation noise to each point of the trajectories, drawn from a centred Gaussian distribution with standard deviation drawn uniformly between 15 and 40 nm (to include the signal intensity dependence of the localisation precision). The time lapse between recordings was set equal to that of the camera (15,4 ms).

The neural network (the architecture of which is detailed below) was then trained to infer two characteristics of interest from the trajectories: their anomalous diffusion exponent (if applicable), and the random walk model from which they were generated among the five described above. Throughout the training, the network processed $\sim 10^6$ independent simulated trajectories.

**Graph neural network and random walks.**   Here, we detail the architecture of the neural network which we use to compute summary statistics vectors describing trajectories. Named GRATIN (for "Graphs on trajectories for inference"), it is inspired from the architecture presented in our previous works [40, 41]. An important feature of this encoder network is that the size of its output is independent of the trajectory length, facilitating the comparison of trajectories of different sizes. Details about the subsequent analyses are found in the next subsections, and we refer the interested reader to the Supporting Information for implementation details.

The code for the encoder network (architecture and training) is available at https://github.com/hippover/gratin.

Graphical models are methods of choice to handle complex inferences [15, 42], model large scale causal relationships [43] and provide inductive biases in Bayesian inferences [44]. Over the last five years, graph-based analysis methods have been complemented by graph neural networks (GNNs), which extend classical neural network approaches to learn and process relations between data points in a flexible manner, using graph-convolutions operations [45]. GNNs are efficient at representation learning [46–48] and naturally apply to data of varying dimensions. Furthermore, GNNs are ideally suited to encode natural symmetries of physical systems [49]. For these reasons, we chose to use a such network to process trajectories.

To do so, we represent a trajectory $\mathbf{R} = (\mathbf{r}_1, \mathbf{r}_2, \ldots, \mathbf{r}_N)$ by a directed graph $G = (V, E, \mathbf{X}, \mathbf{Y})$. Here $V = \{1, 2, \ldots, N\}$ are the nodes, each associated with a position in the trajectory, $E \subseteq \{(i, j)|(i, j) \in V^2\}$ is the set of edges connecting pairs of nodes, $\mathbf{X} = (\mathbf{x}_1^{(0)}, \mathbf{x}_2^{(0)}, \ldots, \mathbf{x}_N^{(0)})$ are node feature vectors, and $\mathbf{Y} = (\mathbf{y}_1^{(0)}, \mathbf{y}_2^{(0)}, \ldots, \mathbf{y}_{|E|}^{(0)})$ are edge features. Each node feature vector $\mathbf{x}_i^{(0)} \in \mathbf{X}$, of size $n_x = 6$, captures information associated to the course of the trajectory until point $i$: normalised time (between 0 and 1), normalised sums of step sizes to the powers 1, 2 and 4, up to point $i$, normalised distance to origin and maximal distance to origin up to point $i$. The normalization terms were chosen so that features do not directly account for the trajectory's scale (a global scale is computed instead and intervenes downstream of graph convolutions). The edges incoming at each node originate only from nodes in the past (respecting causality): node $i$ receives edges from nodes $i - \lfloor \gamma^0 \rfloor, i - \lfloor \gamma^1 \rfloor, \ldots, i - \lfloor \gamma^{k-1} \rfloor$, with $\gamma = (i - 1)^{1/(k-1)}$ and where $k$ is the fixed maximal in-degree of nodes. We remove eventual repeated edges. For trajectories shorter than $k$, we connected each node to all its predecessors. In the following, we chose $k = 10$.

Each edge feature vector $\mathbf{y}_e^{(0)} \in \mathbf{Y}$, of size $n_y = 6$, contains information about the trajectory's course between the two nodes it connects: normalised time difference, normalised distance, normalised sums of step sizes to the powers 1, 2 and 4, between points $i$ and $j$, correlation of the two displacements. The graph construction is illustrated in Fig 2A and exact expressions of edge and node features are provided in S1 Text.

After initialization, the nodes and edges features are embedded into a space of higher dimension (16 for edges, 10 for nodes) using a multi-layer perceptron. Then, nodes vectors are sequentially updated by a series of graph convolution layers, as illustrated in Fig 2B. More precisely, we used three successive GIN convolution layers introduced in [50], with 32 convolution filters each. A pooling layer follows these convolutions, aggregating all node features vectors of a trajectory using an attention mechanism, such that in the second part of the network, each trajectory is represented by a single vector whose dimension is independent of the trajectory length. A feature accounting for the scale of the trajectory is concatenated to this vector, which is then passed to a multi-layer perceptron. Its output is a 16-dimensional vector of learnt summary statistics, which we designate in the following as the "latent representation", or "latent vector" obtained from GRATIN. We purposely chose to use a relatively high number of latent dimensions to benefit from the effects of over-parameterization [51]. During the training phase, the latent vector is fed to two separate MLPs predicting the anomalous exponent and the underlying model of the random walk, respectively. The loss minimised during the network's training is the sum of two task-specific losses: the mean squared error of the prediction of the anomalous exponent (excluding OU trajectories which are not anomalous random walks) and the cross-entropy of the predicted and true model classes. Training the network on such physically informed tasks makes it build a relevant latent representation of

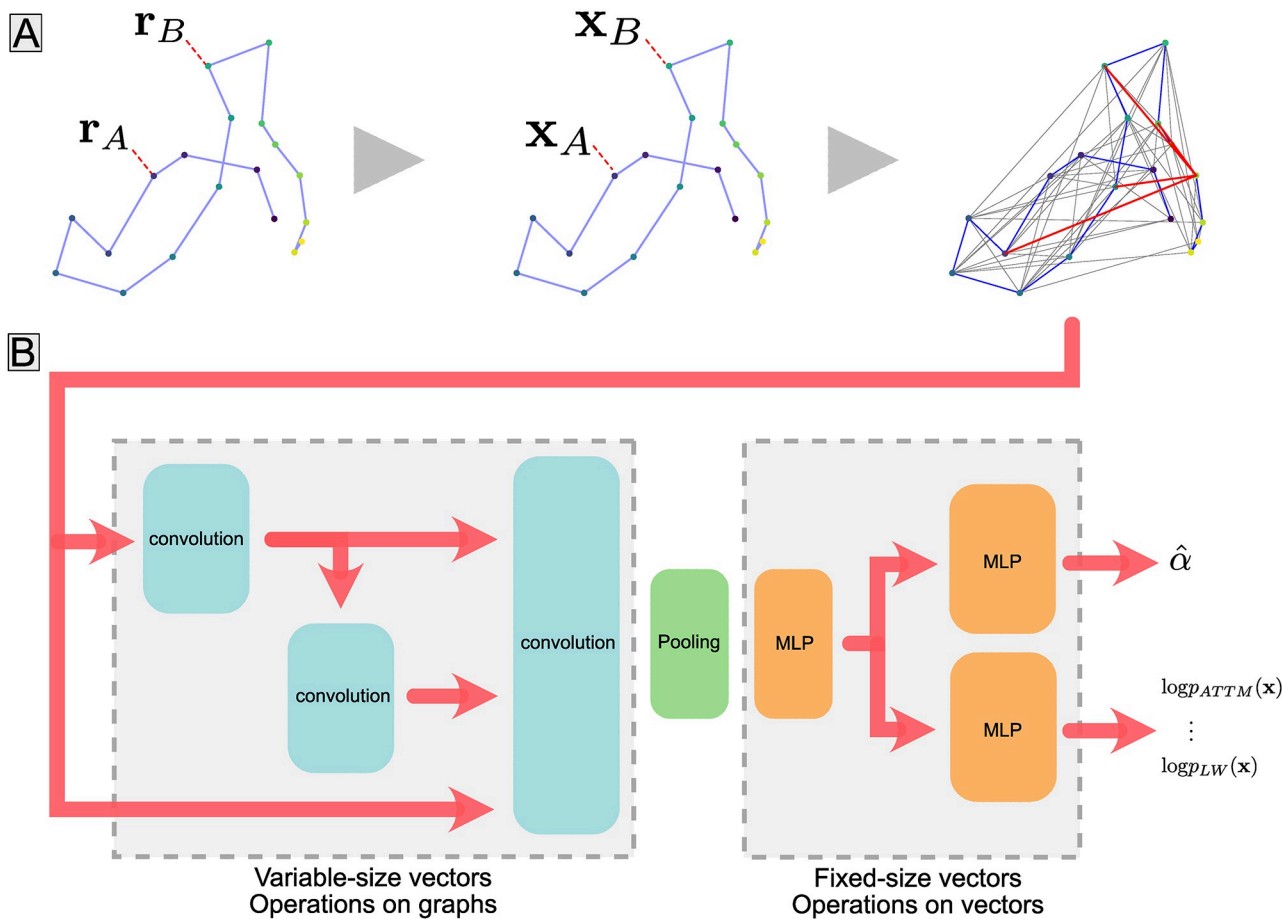

**Fig 2. Model architecture.** (**A**) Construction of a graph from a single trajectory (left). Positions, colored according to time, are treated as nodes for which features **X** are computed (middle). Nodes are then connected by edges (grey lines) following a given wiring scheme, and edge features **Y** are computed. Edges terminating at the trajectory's last point are shown in red. Feature matrices for both nodes (**X**) and edges (**Y**) are depicted in small insets with color coded values. (**B**) Graph neural network. The graph is passed through a series of graph convolution layers (shown in blue), which propagate information along edges. The pooling operation (green) combines all node feature vectors from a graph into a vector of fixed size representing the graph. This vector is then passed to a multi-layer perceptron (MLP in orange), whose output we refer to as the "latent representation" of the trajectory. The latent representation is fed to two task-specific MLPs: one that predicts the trajectory's anomalous exponent $\alpha$ and one that assigns a vector of probabilities for the trajectory to have been generated by each of the models considered.

trajectories, as shown in our previous work [40]—the architecture presented here is an improved version of the previously published one.

**Latent representation of trajectories.** As mentioned above, once processed by the encoder, each trajectory is projected into a 16-dimensional space. Maintaining a relatively high dimension for the latent space helps the network's convergence. Nonetheless, such high-dimensional statistics vectors are not convenient for visual interpretation of the subsequent results. We thus chose, in the rest of the process, to resort to a compressed 2D version of this latent vector, projecting it on a plane using parametric-UMAP [52]. This variation of UMAP allows us to learn the transformation projecting the data from 16 to 2 dimensions on simulated trajectories, so that it is independent of the experimental trajectories and is only trained once. The GNN was trained first, then the parametric-UMAP projection was learnt and both their sets of weights were frozen. We designate as "GRATIN 2D latent representation" the two-dimensional vector, output by the parametric-UMAP for each trajectory. We compared the

power of MMD tests based on the 16- and 2-dimensional latent representations of trajectories and found no significant difference of performance (see S4 Fig and S1 Text).

By comparing Fig 3A, 3B and 3C with respectively Fig 3D, 3E and 3F, we first see that latent representations of simulated and experimental data largely overlap, and that the experimental trajectories fall within the region covered by the simulated ones. We furthermore observe that the random walk model, the diffusivity and the anomalous diffusion exponent are prominent determinants of the latent space structure. Finally, the fact that parameter values predicted on experimental trajectories match the true values of simulated trajectories lying in the same region of the latent space demonstrates the robustness of the encoding. Fig 3G, 3H and 3I illustrate the diversity of $\alpha$Syn:Eos4 trajectories that can be found in a presynaptic bouton, and how this diversity is captured by the latent representation: trajectories whose latent vectors are located in remote regions of the latent space exhibit distinct dynamics. This diversity of dynamics within a given synapse suggests that $\alpha$-synuclein molecules can transition between various dynamic modes and that its state is not exclusively determined by its location.

Using the approach described in this subsection, we can associate to any set of trajectories, a set of constant-sized vectors characterising their dynamics. Each microscope recording, or organelle within it, can thus be characterised by a set of $N$ 2-dimensional feature vectors, $N$ being the number of trajectories. The conditions are thus met to perform statistical testing.

As already stated in introduction, we have shown that this description of trajectories is robust to noise and short trajectory length, and accounts for several aspects of the dynamics, but it is entirely possible to follow the process described in the following paragraphs with another set of descriptive statistics if the features on which the analysis should be focused are well circumscribed. We verified that latent spaces obtained with models trained on different tasks and random walk models were exploitable as well, and hence that the results presented hereafter are not too dependent on this precise method of training. Results are shown in S5 Fig.

## Statistical testing between sets of descriptive vectors

We develop in this section the statistical test we use to compare the dynamics of sets of trajectories, here corresponding to recordings of single molecules observed either in different organelles, fields of view, or under different biological conditions. Each set of trajectories is characterised by the set of corresponding descriptive vectors (one vector per trajectory). Thus, we base our statistical test on the comparison of the generating distributions of these vectors. In the absence of *a priori* knowledge of these distributions, we employ a kernel-based approach: the MMD test.

**Maximum mean discrepancy.**   Maximum mean discrepancy (MMD), introduced in [14], is a measure of distance between distributions. It was developed to perform statistical testing between two sets of independent observations lying in a metric space $\mathcal{X}$, $X = \{x_1, \ldots, x_m\}$ drawn from probability measure $p$ and $Y = \{y_1, \ldots, y_n\}$ drawn from $q$, with the the goal of assessing whether or not $p$ and $q$ are different.

Given a class $\mathcal{F}$ of functions from $\mathcal{X}$ to $\mathbb{R}$, the MMD between two probability measures $p$ and $q$ is defined as

$$\text{MMD}[\mathcal{F}, p, g] = \sup_{f \in \mathcal{F}} \left( \text{E}_x[f(x)] - \text{E}_y[f(y)] \right), \tag{1}$$

where $\text{E}_x$ and $\text{E}_y$ denote expectation w.r.t. $p$ and $q$, respectively.

If the function class is the unit ball in a Reproducing Kernel Hilbert Space (RKHS) [14] $\mathcal{H}$, the square of the MMD can directly be estimated from data samples. Denoting $k$ the kernel operator such that $\forall f \in \mathcal{F}, f(x) = \langle f, k(x, \cdot) \rangle$, an unbiased estimator of the square of the

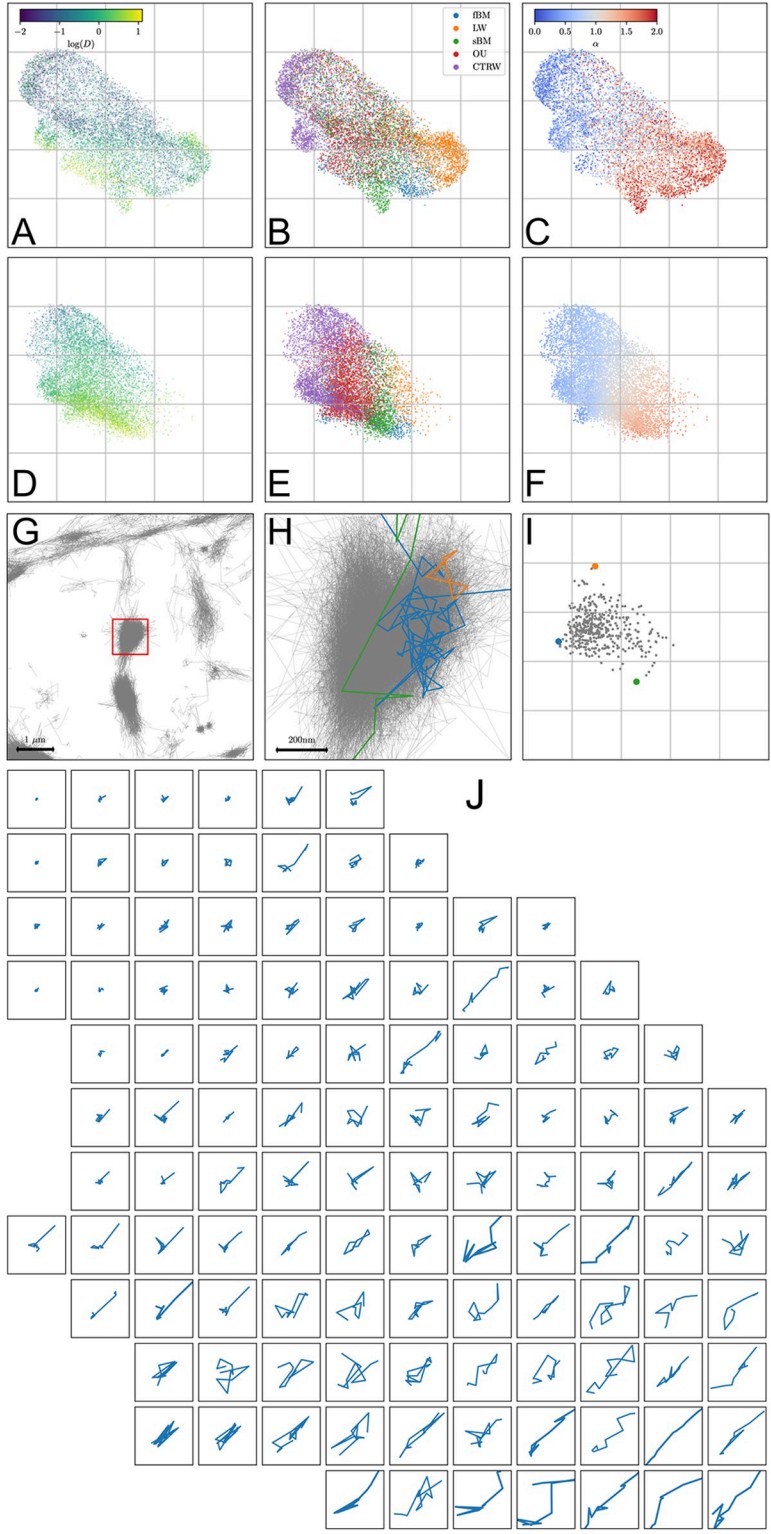

**Fig 3. 2D Latent space of trajectories. (A, B, C)** Latent representations of simulated trajectories, colored according to the true log-diffusivity, the true random walk model (fBM: fractional Brownian motion; LW: Levy walk; sBM: scaled Brownian motion; OU: Ornstein-Uhlenbeck process; CTRW: continuous-time random walk) and the true anomalous diffusion exponent. **(D, E, F)** Latent representations of recorded trajectories, colored according to the same variables as above, this time using inferred values. **(G)** Recorded trajectories at synapses and in the axon. **(G)** Zoom on a presynaptic bouton, delimited by the red square in panel D. Three individual trajectories are highlighted. **(I)** Latent

representation of the trajectories at this synapse, with colored dots corresponding to the three trajectories highlighted in H. (**J**) Examples of acquired trajectories, located according to their position in the latent space. Each square has a side length of 1.5 micrometer, trajectories length vary from 14 to 16.

MMD between $X$ and $Y$ is given by:

$$\mathrm{MMD}_u^2[\mathcal{F}, X, Y] = \frac{1}{m(m-1)} \sum_{\substack{i,j \\ i \neq j}} k(x_i, x_j) + \frac{1}{n(n-1)} \sum_{\substack{i,j \\ i \neq j}} k(y_i, y_j) - \frac{2}{nm} \sum_{i,j} k(x_i, y_j) \tag{2}$$

In our case, $\mathcal{X} = \mathbb{R}^2$, and we used the classical Gaussian kernel $k : x, y \rightarrow k(x, y) = \frac{1}{\sqrt{2\pi}\sigma} \exp\left(-\frac{\|x-y\|_2^2}{2\sigma^2}\right)$. We set the kernel bandwidth $\sigma$ either to the median of the pairwise Euclidian distances between samples from X and Y or we optimised it in specific conditions.

The MMD is capable of detecting subtle differences such as the ones between data generated by generative adversarial networks (GANs) and real data [53]. It has also proved efficient in discovering which variables exhibit the greatest difference between data sets [14, 54].

**Statistical test.**   We adapted the bootstrap test described in [14] to assess whether dynamics of two sets of trajectories exhibit significant differences. The sets of trajectories are represented by their two sets $X$ and $Y$ of descriptive vectors, respectively drawn from unknown probability densities $p$ and $q$. For simplicity, we assume here that $X$ and $Y$ have the same number of elements, $m = n$, using the same notation as in the last section. In practice, the number of observed trajectories varies significantly across experimental replicates. Hence, to ensure that all replicates have an equal importance when the two sets do not have the same number of trajectories we randomly sub-sampled the larger of the two sets to equalise their sizes. The null hypothesis $H_0$ of the statistical test is that $p = q$, i.e. the two conditions lead to the same distribution of random walks. Under $H_0$, we approximated the distribution of $\mathrm{MMD}_u^2[\mathcal{F}, X, Y]$ by bootstrapping, i.e. we drew random samples from the union of $X$ and $Y$ and distributed them in two groups $X'$ and $Y'$ (whose sizes respectively match those of $X$ and $Y$), on which we computed $\mathrm{MMD}_u^2[\mathcal{F}, X', Y']$. We repeated this procedure a sufficient number of times to obtain an estimation of the distribution of $\mathrm{MMD}_u^2$ under the assumption that $X'$ and $Y'$ are drawn from the same distribution. Then, if the original $\mathrm{MMD}_u^2[\mathcal{F}, X, Y]$ was greater than the $1 - \beta$ quantile of this distribution, we rejected $H_0$. This test is said to be of level $\beta$, because with probability $\beta$, we will reject the null hypothesis when it is actually true.

Some factors unrelated to the dynamics of a biomolecule might have an influence on its descriptive statistics: this is notably the case of the trajectory length, or of the localization uncertainty. To compensate these undesired differences, it is possible to "marginalize" for these factors prior to comparing two sets $X$ and $Y$ of descriptive vectors, by making sure that the distribution of such factors is equal in both sets. In the following, we always marginalized comparisons by trajectory length. In cases when a large number of conditions are analyzed, and if one is worried that p-values of comparisons might be close to the significance threshold (because of a low number of trajectories, or when the difference between conditions is expected to be subtle), we advise to carefully choose the comparisons to perform, in accordance with pre-existing biological knowledge, so as to limit the amplitude of the correction to apply to the significance threshold (e.g. using the Bonferroni method).

## Results

### Detecting differences of diverse nature between sets of short simulated trajectories

To assess more precisely the performance of the statistical testing framework based on 2-dimensional GRATIN vectors, we applied it on simulated data. We set the threshold of statistical significance to $\beta = 0.05$, and we simulated trajectories as described in Material and Methods.

A first case of our test is to detect changes in the proportions of given types of trajectories between two sets of observations. This is illustrated in Fig 4, where we show example comparisons in the 2D latent space between two sets with different proportions of their trajectories generated by fBM and sBM. We compared fBM and sBM, since they share numerous features and because for a large range of values of the anomalous exponent they are challenging to distinguish [40]. Furthermore, these two random walk models are highly representative of our experimental data, as can be seen by their latent space occupations (compare Fig 3A and 3B).

The difficulty of separating the two populations depends on their relative proportions in the two datasets, and we see that both the amplitude of the witness function (Fig 4C) as well as the value of the test statistic (Fig 4D) decrease as the ratio is closer to 1:1. When the two sets are drawn from the same 50/50 distribution, the test does not, and should not, find significant differences between them.

The other main factor determining the difficulty of detecting a difference, is the size of the data sets. Experimentally, changes in biological conditions lead not only to changes in the properties of the random walks. It also leads to changes in the total number of trajectories of a given type. This causes challenges in performing proper statistical testing. To quantitatively assess the effect of both the number of trajectories and the relative proportions belong to different random walk classes, we conducted numerical experiments where we varied these two parameters systematically (S2 Fig (A)).

Besides differences in the proportions of trajectories generated by different random walk models, the sets may also differ in the models' parameter values. We thus additionally evaluated the test's ability to distinguish two sets of fBMs, one with anomalous diffusion exponent $\alpha = 1 - \delta$ and the other with $\alpha = 1 + \delta$. Our results indicate that in both cases 1 000 trajectories are sufficient to detect subtle changes between distributions (S2 Fig). Fewer trajectories are needed to detect starker differences. In cases where the compared sets are drawn from the same distribution ($v = 0$ and $\delta = 0$), the null hypothesis is rejected in about 5% of cases, consistent with our chosen $\beta$-level. It is worth noting that sBM and fBM—which our method distinguishes well even in the low sample size limit—are "unequal twins", borrowing the expression of [55], since they share the same marginal probability distribution function despite being very different processes. We show in S4 Fig that the MMD test is, in most cases, more powerful than Hotelling tests, and that the GRATIN 2D features allow a better separation of random walk models. The details of these experiments are provided in S1 Text.

One way to further improve these results is to optimise the kernel used to compute the MMD. We show in S1 Fig how kernel bandwidth and shape affect the power of the test. If the kernel bandwidth is too small, this weakens the test by making it too sensitive to noise. Conversely, if its bandwidth is too large, this prevents the test from detecting subtle changes. We tested the effect of the kernel characteristics in the same setting as illustrated in Fig 4 and S2 Fig (A), with $N = 200$ trajectories in each set and comparing sets with 70% fBM / 30% sBM and 30% fBM / 70% sBM. We observed that a Gaussian kernel with radius $\sigma$ equal to the median pairwise distance in the dataset (i.e. $\sigma$ between 1.5 and 2) yields a near-optimal test, in agreement with earlier findings [14]. Finally, while we have here focused on optimizing type II error

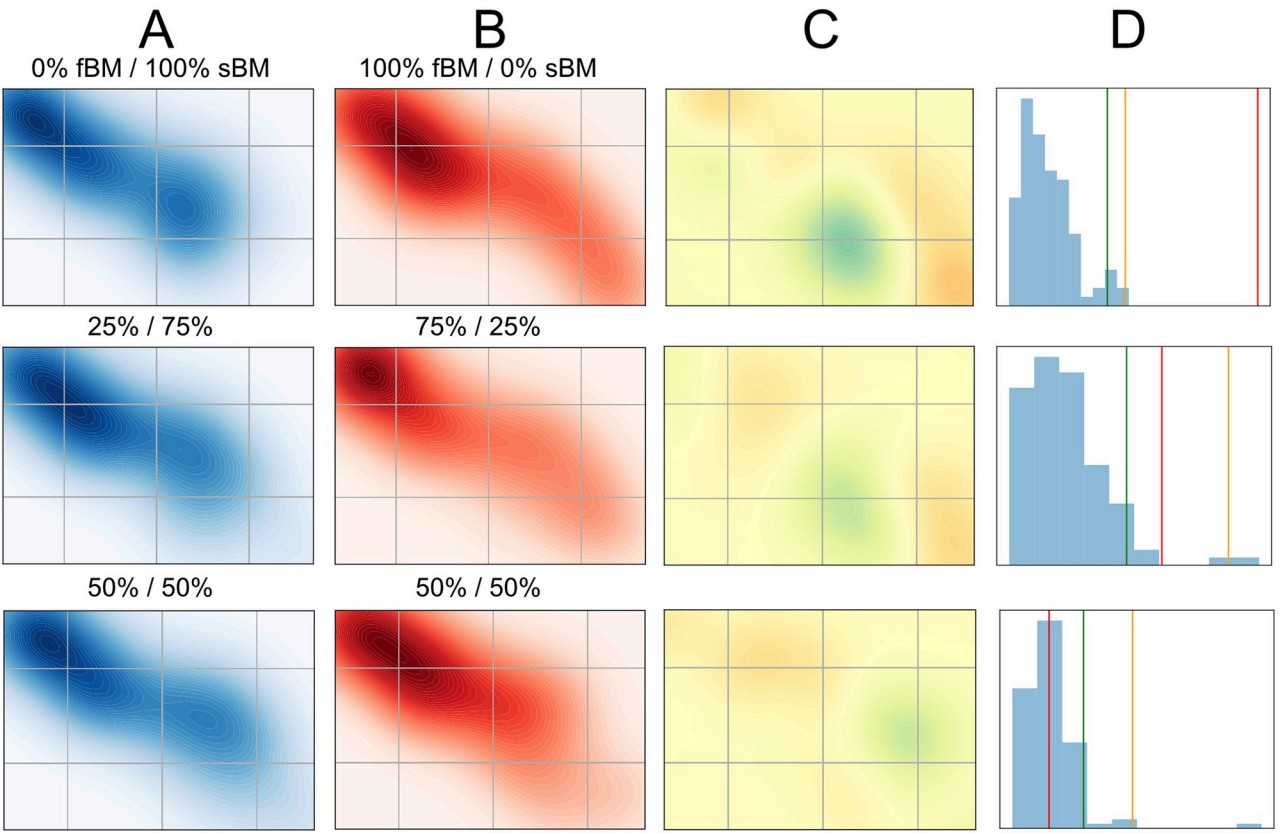

**Fig 4. MMD-based statistical test.** (**A**), (**B**) Densities of latent vectors in the 2D plane, for two sets of 500 trajectories with different ratios of fBM / sBM trajectories (top: 0% fBM / 100% sBM vs. 100% fBM / 0% sBM, middle: 25% / 75% vs. 75% / 25%, bottom: 50% / 50%). (**C**) Witness functions of the MMD test for difference between A and B, i.e. the function attaining the maximum in Eq 1, based on the available samples. (**D**) Distribution of the test statistic $\mathrm{MMD}_u^2$ between sets of equal size composed of randomly chosen trajectories of the two sets, with its top 1% (yellow line) and top 5% percentiles (green), as well as the unbiased estimate of the square MMD between the two sets (red).

while controlling type I error (i.e. fixed $\beta$-level), the parameters and the functional form of the kernel can be adjusted to control either type I or II error while optimising the other [56].

### Differences of $\alpha$-synuclein mobility in axons and at synapses in response to membrane depolarisation

We analysed trajectories of $\alpha$Syn:Eos4 molecules in the axons of cultured cortical neurons and compared them between different subcellular regions, outside or inside the synaptic bouton, and experimental conditions, control, high KCl (leading to synaptic depolarisation) and fixed cells.

The three experimental conditions (control, KCl, fixed) and two subcellular regions (extra- and intra-synaptic) define six populations of trajectories, whose latent space occupation densities are shown in Fig 5A and 5B. In the remaining four columns of Fig 5, we illustrate a few comparisons performed between pairs of trajectory populations using our statistical test. Fig 5C and 5D show the latent space densities for the two conditions that are compared in each case, the differences of which are the witness functions shown in Fig 5E. The distributions of the test statistics under the null hypothesis, obtained after 1 000 bootstrapping iterations, are shown in panel F, as well as its top 1% and 5% quantiles and the test statistic obtained on the actual two compared populations.

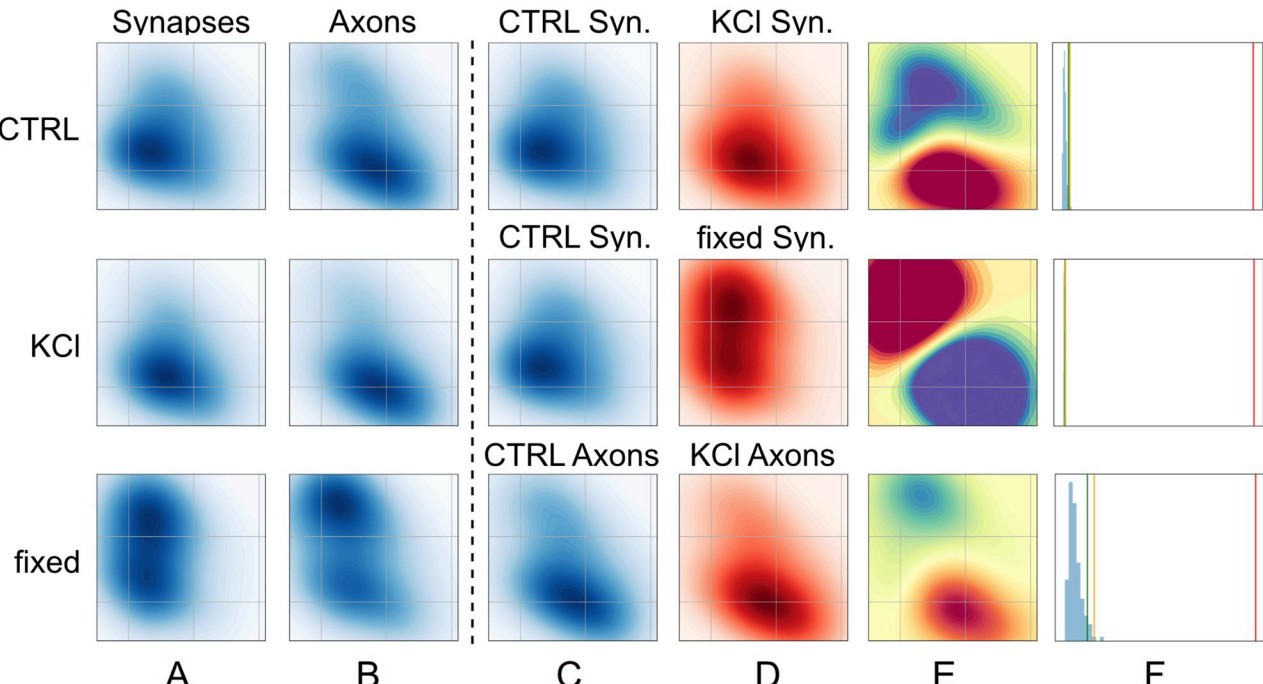

**Fig 5. Latent space occupation & statistical testing.** Left part: (**A**), (**B**) Latent space occupation densities of $\alpha$Syn:Eos4 trajectories observed in synapses and axons, in the three experimental conditions (control, high KCl, fixed). Right part: each row is one comparison of two sets of trajectories. (**C**), (**D**) Side by side comparison of the latent space occupation densities of the two sets of trajectories used in the comparison in E and F. (**E**) Witness function of the comparison. The colour scale is preserved across rows. (**F**) Histograms illustrating the statistical test based on the MMD. The green and yellow vertical lines represent the top 5% and top 1% quantiles of the null distribution of the squared MMD, respectively, while the red line shows the squared MMD for the experimental data.

In all of the three cases illustrated in Fig 5, the empirical $\mathrm{MMD}^2_u$ is significantly higher than the top 1% quantile of the null distribution, meaning that our test detects a significant difference in the properties of the trajectories of the two compared subsets at the 1% $\beta$-level. Note the wide range of magnitudes spanned by these differences, which is well judged by looking at the absolute intensity of the witness functions shown in Fig 5E. We see that the difference induced by fixation on $\alpha$-synuclein mobility at synapses (Fig 5E, middle) is much more pronounced than the one induced by high KCl treatment (Fig 5E, top). This is also apparent when looking at how large $\mathrm{MMD}^2_u$ is compared to its distribution under the null hypothesis: in the case of the fixed vs control comparison (Fig 5F, middle), the histogram of the $\mathrm{MMD}^2_u$ values obtained under the null hypothesis is completely squeezed to the left because the empirical $\mathrm{MMD}^2_u$ is more than two orders of magnitude larger than the top 1% quantile of the distribution under the null hypothesis. In comparison, KCl treatment produces a less drastic change in $\alpha$-synuclein trajectories located in synaptic terminals (Fig 5F, top), although this effect is also highly statistically significant ($p \ll 0.01$).

We further observed that, while their magnitudes differ, the witness functions of the control/KCl comparisons in axons and in synaptic boutons (Fig 5E, top and bottom) exhibit similar patterns. This indicates that the addition of KCl to the medium can affect the physical properties of many if not all $\alpha$-synuclein molecules in a similar manner, irrespective of their sub-cellular location. In contrast, $\alpha$-synuclein mobility in fixed neurons appears to be almost entirely abolished, which is seen not only in the amplitude of the change, but also in the fact that the occupation of the latent space displays massive qualitative differences in this

condition. This demonstrates that $\alpha$-synuclein is highly mobile in living cells, and helps to put our experimental findings into perspective.

An interesting feature of the MMD, is the possibility of extracting the points of the feature space that are most important for distinguishing one distribution from another. By finding the local maxima of the $S$ statistic, introduced in [57], which is given as the ratio of the mean squared amplitude and the variance of the witness function estimated by bootstrapping, we identify the regions of the latent space where the occupation differs most in the two populations. This enables a straightforward interpretation of the latent space. In Fig 6, we apply this method to the comparison of intra-synaptic $\alpha$Syn:Eos4 trajectories in the control and KCl conditions. Fig 6A and 6B show the latent space occupation densities of these two groups of trajectories. Fig 6C shows the witness function, and Fig 6D the variations of the $S$ statistics across the latent space. The "critical region", where $S$ is close to its maximum (we used as threshold 3/4 of the maximum), defines a type of trajectories which are maximally discriminative between the two compared conditions. Here, this region overlaps with the area of the latent space where trajectories from the KCl condition predominate over those from the control condition. Another critical region can be defined around the maximum of $S$ in the region where trajectories from the control condition predominate. Example trajectories from each of these two critical regions are shown in Fig 6E and 6F. Besides, we show in Fig 6G–6K the histograms of several descriptive quantities, for each condition and each condition's critical region. We see on these histograms that the separation is in general much clearer when restricting to trajectories from critical regions, something which further validates the encoding of relevant physical quantities in the latent space.

According to this analysis, the representative $\alpha$-synuclein trajectories exhibit a greater mobility in the depolarised state, and a more subdiffusive motion in the control condition. This is likely the result of a weaker binding of $\alpha$-synuclein at synapses, as reflected in the overall reduction of $\alpha$-synuclein molecules during KCl application (Fig 1A and 1B). The observed difference in asymmetry is likely the trace of trajectories entering/leaving the synapse more frequently in the depolarised state (we recall that trajectories were considered "in synapse" when more than half their localizations are inside a same synapse).

Furthermore, as illustrated in S3 Fig, we check that all acquired fields of view contribute evenly to the difference between the control and KCl conditions. To do so, we compared the proportion of trajectories coming from each field of view and condition in the main critical region. We could thus confirm that trajectories located in this region of the latent space originate from all considered fields of view in a balanced manner, and that within each field of view, the difference of representation of each condition is in the same direction (except for one field of view, where there is almost no difference). This furthermore excludes the possibility that the observed differences are solely due to a single abnormal recording.

## Comparing synapses and biological replicates

The MMD permutation tests whether two groups of trajectories are observations of a same stochastic process. Besides, the MMD provides a measure of distance between probability densities, and the magnitude of such distances can be compared to assess the relative "proximity" of groups. Groups may be defined according to any relevant level of granularity provided that each contains enough trajectories to limit the variance of MMD estimates (a few hundreds of trajectories is usually sufficient), offering a powerful mean of probing an experimental dataset's heterogeneity, not restricted to inter-condition comparisons. Here, we give an example of such possibility, by grouping trajectories by synapse and computing the value of $MMD_u^2$ between all pairs of synapses. We obtain an inter-synapse distance matrix, shown in Fig 7A. Using these

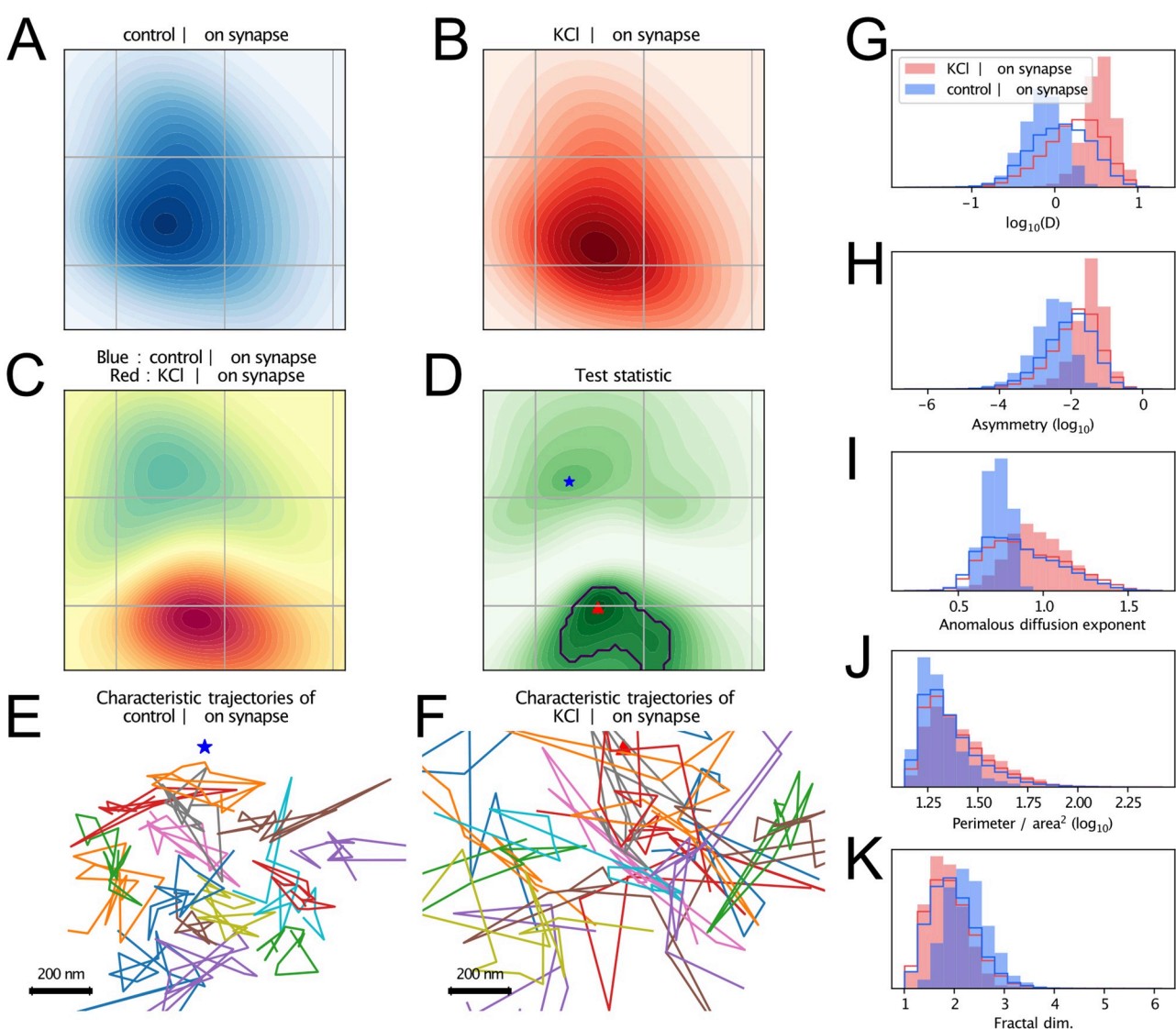

**Fig 6. Most salient dynamics.** (**C**) Witness function of the comparison of intra-synaptic trajectories, between the control (blue) and KCl (red) conditions. (**D**) Test statistic such as defined in [57], i.e. ratio of the square amplitude divided by the variance. The black contour indicates the "critical region" of the (KCl, on synapse) condition, i.e. the region of the latent space where this condition predominates and which is most responsible of the statistical difference. (**E**) and (**F**) Illustration, for each maximum of the test statistic, of its 16 closest trajectories. (**G-K**) Histograms of descriptive quantities, grouped by conditions. Outlined histograms correspond to the whole population of trajectories of a condition, while filled histograms correspond only to those located in the "critical" region of each condition (hence the better separation).

distances, we can embed the trajectory subsets in an Euclidian space, i.e. summarise each subset by a vector of fixed dimension, using for instance the multi-dimensional scaling (MDS) algorithm [15]. We adapted the MDS algorithm in order to account for the uncertainty that we have in the estimation of the squared distance, which notably depends on the number of observed trajectories per synapse (see S1 Text). We show in Fig 7B the vectors obtained when using this method to embed synapses in a three-dimensional space. The clouds of points corresponding to synapses observed in each condition do not entirely overlap, which is expected given that we previously showed that conditions were significantly different. This visualization complements the statistical test by providing an intuitive illustration of the relative extent of

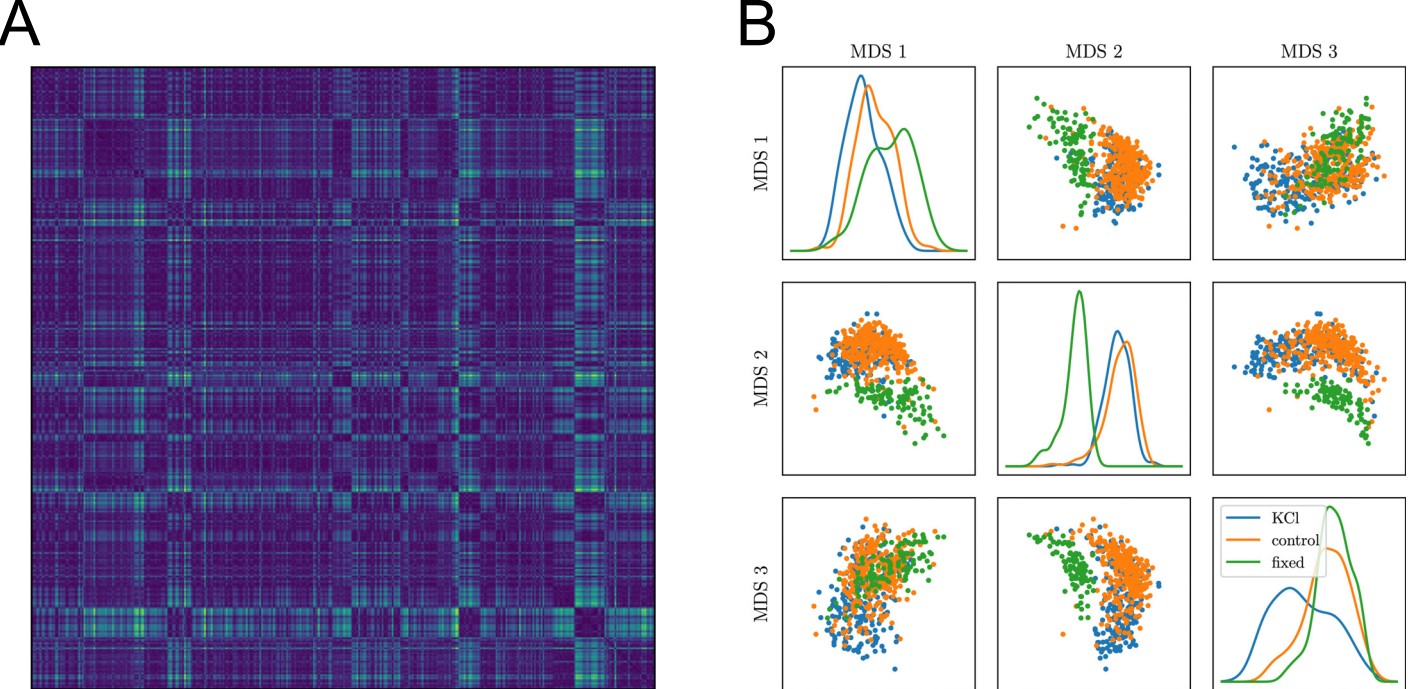

**Fig 7. Comparing individual synapses.** (**A**) Matrix of the inter-synapse MMD: the value at row $i$ and column $j$ is colored according to the MMD between trajectories of synapse $i$ and synapse $j$. (**B**) three-dimensional embedding obtained by multi-dimensional scaling (MDS) based on this distance matrix. Dots are coloured according to the condition in which the synapse they represent was observed: CTRL in orange, KCl in blue and fixed in green.

inter- and intra-condition variability. More generally, this type of visualisation may be used to highlight potential continuous or discrete degrees of freedom of the observed system, resulting in connected or disjoint point clouds.

## Discussion

We have introduced a statistical procedure to compare organelles or biological conditions of single molecule experiments. This statistical test does not require explicitly determining the generative models of experimentally recorded random walks. The test consists of two steps. In the first step, an amortised inference is used to reduce any trajectory to an optimized feature vector of constant size. In the second, the distribution of features between different conditions is compared using the MMD statistical test.

Our approach allows extracting physically and biologically relevant results without having to assign the biomolecule motion to canonical models. Although such models are instrumental for interpreting the properties of random walks, the complexity and heterogeneity of biological environments at the nanometer/micrometer scale often precludes unambiguous model assignation.

An approach that is conceptually related to the framework proposed here is Bayesian model averaging (BMA). BMA does not look for a single best model to describe experimental data but rather evaluates the posterior distributions for each of the possible models and then averages the different results. It is challenging to apply BMA to our current problem for two reasons: (i) the space parameters of the different random walks are not identical and (ii) the evaluation of the posterior distributions associated with each random walk by Bayesian methods is not possible for all models. It would be possible to develop variational methods to treat

models that do not have tractable likelihoods. However, the marginalisation of the different models would be computationally intensive. Moreover, even though BMA could include information from multiple models, it would still rely on the projection of the experimental date onto a set of canonical models, whereas our framework can interpolate between the models.

We applied our approach to study the dynamics of $\alpha$-synuclein molecules in axons and presynaptic boutons. In agreement with earlier studies of the population dynamics of $\alpha$-synuclein [23, 24], we found that the protein assumes differing dynamic states at synapses and in axons. Depolarisation of the presynaptic terminals through the application of high potassium concentrations shifted the relative frequency of the various states, without necessarily changing the types of diffusion. In other words, our analysis highlights that the proportion of $\alpha$-synuclein proteins found in a mobile state (superdiffusive, with a high diffusion coefficient) is higher after depolarisation.

Our statistical testing framework paves the way to automated analysis of single molecule experiments. Single molecule pharmacology is an emerging field [58, 59], in which the effects of drugs are evaluated at the nanometer scale by studying the spatial properties and dynamics of biomolecules of interest. The possibility to automatically compare different conditions without relying on manually selected generative models of molecule diffusion would be helpful in defining groups of conditions in which a certain effect can be detected. Even though model identification will often be impossible, the properties of the latent space can reveal the source of observed differences. The witness function can thus be instrumental in differentiating changes in the probability of occupancy of specified domains within the latent space between conditions. As illustrated on Fig 6, going from a region of the latent space to an intelligible trait of trajectories is rather intuitive, hence the interest of this method to orient further analysis and build biologically relevant hypotheses.

Beyond the automation of the analysis procedure between biological conditions, our approach is well suited for exploratory data analysis. The capacity to project individual, differently sized trajectories into finite sized vectors makes it possible to study precise sub-cellular compartments or organelles in a standardised form, and thus allows to test statistical differences between these regions. Hence, recorded single molecule data can be searched in order to detect and characterise regions of the cell that have different statistical properties. This exploration can be done even in regions with different trajectory densities, as is the case for $\alpha$-synuclein at synapses versus axonal domains. The general idea of using simulation-based inference to learn rich and robust descriptive statistics, on which statistical tests can then be performed, could well be adapted to the analysis of other types of biological data.

One of the current limitations of the current approach is the difficulty in evaluating the type II error [56] bounds on the statistical test. The MMD test is applied within the latent space of the GNN. This manifold is built using a set of non-linear operations, which depends both on the numerical trajectories seen during the training and on the cost function being optimised. Hence, there may be domains within the latent space that could lead to improper sensitivity of the statistical test. As can be seen in [40] and in Fig 3, different types of random walks occupy domains of different size and there is a large overlap of the regions. Since our approach relies on a simulation based framework, it is possible to use numerical simulations matching the experimental occupancy of the latent space to evaluate the accuracy of the test. Furthermore, extensive simulations allow to check if the statistical test misbehaves, even though this procedure can be time consuming. In order to further improve the statistical power of the test, one could optimise the kernel with which the MMD is computed. Along these lines, a possible variant of this method could rely on an encoder network trained not on a supervised inference task but rather to maximise the MMD between two sets of

experimentally recorded trajectories. This, however, would require a substantially larger quantity of experimental data.

## Supporting information

**S1 Text.** Details about the MDS with uncertainty, the GNN architecture and training, the comparison with other statistical tests, as well as a study of the influence of the training dataset and supervised task.
(PDF)

**S1 Table.** Sizes of multi-layer perceptrons used in the neural network.
(PDF)

**S1 Fig. Influence of kernel parameters on the test's power.** Probability of rejecting the null hypothesis that the two sets of trajectories are drawn from the same distribution, with varying kernel types and radii. The sets are drawn with the same characteristics as those used for S2 Fig (A) with $N = 200$ and $v = 0.2$.
(EPS)

**S2 Fig. Performance of the statistical test on simulated trajectories.** **A**: Probability of detecting a difference between two sets of $N$ trajectories composed of a fraction $v$ of fractional Brownian motions and $1 - v$ of scaled Brownian motions. **B**: Probability of detecting a difference between two sets of $N$ fractional Brownian motions, one with anomalous diffusion exponent $\alpha = 1 - \delta$ and the other with $\alpha = 1 + \delta$. Probabilities are estimated by performing the test 100 times; at each trial, new trajectories are simulated and bootstrap-estimation of the distribution of $\mathrm{MMD}_u^2$ under the null hypothesis is done using 100 random splits.
(EPS)

**S3 Fig. Origin of trajectories found in the critical region.** These counts were obtained using a set composed of the same number $n = 1,000$ of trajectories from each microscopy recording. We randomly subsampled those who had more intra-synaptic trajectories, and discarded those who had less than 1,000 (hence the column with missing number of KCl trajectories).
(EPS)

**S4 Fig. Comparison of statistical tests.** (**A**): Fraction of comparisons of sets of trajectories from each model pair, and based on each type of descriptive vectors, for which the MMD permutation test has more power than the Hotelling (or $t$–test for one-dimensional vectors). (**B**): Fraction of comparisons for which the test based on descriptive vectors from the corresponding row has more power than when based on descriptive vectors of the column. Red means high, blue means low. For instance, the red top cell means "MMD tests based on alpha yielded lower p-values than those based on log(D)". It is clear from these experiments that vectors output by the neural networks carry more information than the analytical ones ($\log_{10}(D)$, fractal dimension, convex hull, asymmetry). (**C**): Cumulative distribution of the $p$–values yielded by each test when comparing two sets of trajectories generated using the same algorithm (over 500 repetitions of the test with independently generated samples). The red dashed line marks the diagonal.
(EPS)

**S5 Fig. Influence of the training task & data on the learnt latent representation.** (**A**): Latent representations of $10^5$ trajectories of length 15, obtained with models trained on various types of random walks. Left: CTRW and LW, middle: sBM only, right: fBM only. In the top row, dots are colored according to the generative model of their trajectory; in the bottom row, they

are colored according to its anomalous exponent $\alpha$ (blue: subdiffusive, red: superdiffusive). (**B**): Same logic, but with a fixed set of random walk types used for training. This time, we varied the task on which the network was trained: inference of both the anomalous exponent and the type of random walk (left), inference of the anomalous exponent only (middle) or of the type of random walk only (right).
(EPS)

## Acknowledgments

We thank Alexandre Blanc, Michael Lelek, Sylvain Prigent, Mohamed El Beheiry, Srini Turaga, Raphael Voituriez & Bassam Hajj for helpful discussions. Fumihiro Niwa is thanked for technical help.

## Author Contributions

**Conceptualization:** Hippolyte Verdier, François Laurent, Alhassan Cassé, Christian L. Vestergaard, Christian G. Specht, Jean-Baptiste Masson.

**Data curation:** Hippolyte Verdier, Christian G. Specht.

**Formal analysis:** Hippolyte Verdier, Christian L. Vestergaard, Jean-Baptiste Masson.

**Funding acquisition:** Christian G. Specht, Jean-Baptiste Masson.

**Investigation:** Hippolyte Verdier, Christian L. Vestergaard, Christian G. Specht, Jean-Baptiste Masson.

**Methodology:** Hippolyte Verdier, François Laurent, Christian L. Vestergaard, Christian G. Specht, Jean-Baptiste Masson.

**Project administration:** Jean-Baptiste Masson.

**Resources:** Christian G. Specht, Jean-Baptiste Masson.

**Software:** Hippolyte Verdier, François Laurent.

**Supervision:** Alhassan Cassé, Christian L. Vestergaard, Christian G. Specht, Jean-Baptiste Masson.

**Visualization:** Hippolyte Verdier, François Laurent, Christian G. Specht.

**Writing – original draft:** Hippolyte Verdier, Christian L. Vestergaard, Christian G. Specht, Jean-Baptiste Masson.

**Writing – review & editing:** Hippolyte Verdier, Christian L. Vestergaard, Christian G. Specht, Jean-Baptiste Masson.

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
