## [Decision Letter · Decision Letter 0]

25 May 2022

Dear Mr. Verdier,

Thank you very much for submitting your manuscript "A maximum mean discrepancy approach reveals subtle changes in α-synuclein dynamics" for consideration at PLOS Computational Biology.

As with all papers reviewed by the journal, your manuscript was reviewed by members of the editorial board and by several independent reviewers. Both reviewers raise major concerns that should be carefully addressed. In light of the reviews (below this email), we would like to invite the resubmission of a substantially revised version that takes into account all comments from both reviewers.

We cannot make any decision about publication until we have seen the revised manuscript and your response to the reviewers' comments. Your revised manuscript is also likely to be sent to reviewers for further evaluation.

Sincerely,

Bert L. de Groot

Associate Editor

PLOS Computational Biology

Arne Elofsson

Deputy Editor

PLOS Computational Biology

Reviewer's Responses to Questions

**Comments to the Authors:**

Reviewer #1: Verdier et al describe a methodology for comparing sets of biomolecule trajectories as obtained in imaging experiments to identify changes in the associated dynamics. They propose a two-step scheme, involving feature extraction through machine learning (via a graph neural network) and a classical statistics test (maximum mean discrepancy). They apply this approach to detect differences in the dynamics of a presynaptic protein.

The task of extracting information from short and noisy trajectory is relevant and has been included in a recent competition in which the authors took part but they do not mention/cite (Muñoz-Gil et at, Nat Comms 2021). Unsupervised methods have also been recently proposed for this task (Muñoz-Gil JPhysA 54 504001 2021).

The method they present in the manuscript builds on the one they developed for the competition and is based on graph neural networks (Verdier et al, JPhysA – ref 24). Recently, graph neural networks are getting a lot of attention and might potentially be a good strategy for this kind of problem. The results presented in the manuscript seem promising but are mostly presented at a qualitative level and lack comparison with a gold standard. It is difficult to understand the performance of the methods since very little quantitative assessment is presented.

I have a few major concerns:

1. Features are extracted at the single-trajectory level, but the comparison is done between sets of (short) trajectories. Is it really necessary to use such a complex architecture for this task? Wouldn’t other classical approaches at the ensemble level perform similarly (i.e., comparing the distribution of D, or of displacements, or of gyration radii)? For example, Fig 6 suggests that differences are mainly due to a change in mobility (as also stated in the intro) that could be detected also with other methods.

2. The manuscript lacks a baseline with a classical approach and a comparison with other methods designed to tackle similar problems (e.g., the winners of the competition mentioned above). For example, it would be important to understand what the advantage (if any) is of using a graph neural network over a CNN or RNN for this task. This point should be demonstrated by comparing the performance of different architectures on the same set of data.

3. The manuscript lacks an ablation study that is necessary to conclude that the method is performing well due to its good design rather than to the potential artifacts.

4. The architecture requires a more detailed description. It is not clear which features are passed to each node/edge (Only coordinates? Are they normalized? Time? Do they use any embedding?) nor how the features are updated (message-passing?) and aggregated. Figure 2 seems to suggest that the graph is fully connected, but I could not find any mention of this in the manuscript. Is this the case or any sparsification is applied?

5. The link to the code directs to a private repo. The reviewers must be enabled to check the code to better understand the method.

6. Does feature extraction provides any interpretability besides a hint about the generative model? A representation of the UMAP features with respect to the anomalous diffusion exponent (instead of D) might be helpful in this sense.

7. Linked to the previous comment: The additional statistical test somewhat subdues the elegance of an end-to-end trainable model. Is it really necessary? In the end, the results are based on a “soft” classification. In an equivalent fashion, one might think to perform a regression and/or a classification tout court and then just compare the proportion of trajectories in each class.

8. Can’t similar results be directly obtained by extracting global features from the graph neural network?

9. Many references miss journal, page, and year. Relevant literature about graph neural networks is missing (e.g., arXiv:1806.01261)

Reviewer #2: In their manuscript the authors introduce a two-step statistical testing scheme combining simulation-based inference to train a neural network and a maximum mean discrepancy statistical test on the vectors of learned features to compare the features. They characterize sets of simulated random walks and analyze experimental alpha-synuclein traces in synapses in cultured cortical neurons in response to membrane depolarization. The authors also identify the domains in the latent space where the differences between biological conditions are the most significant,

Technically the work seems to be interesting and sound. There are, however, serious conceptual flaws in the motivation, (alleged) state of the art, and even the basic logic of their reasoning (for details please see below).

Several crucial (and recurring) statements are very puzzling, e.g. “… to detect changes in biomolecule dynamics within organelles without needing to identify a model of their motion”. I cannot identify any reason whatsoever why it should be required to identify a model of motion in order to detect differences. Yet, this seems to be the main motivation for the work.

Specific remarks:

1) Basically the authors argue that instead of “describing trajectories using a set of explicitly defined features” (i.e. physics based observables) it is preferable to use a black box. The entire reasoning seems to be based on the premise that one is required to identify some underlying model for the dynamics in order to be able to identify and quantify changes in the dynamics, which is certainly not true. To spell this out, there is in fact no need for an underlying model if one sets out to quantify the properties of trajectories or to detect and quantify differences between them. There is a very large (and I claim open) set of physical observables one may infer directly from trajectory data in a model free fashion. For example, the “canonical” time averaged squared displacement, position power spectra, van Hove functions and single-trajectory van Hove functionals, occupation measures, asphericity of individual trajectories, spectral and fractal dimensions of trajectories, memory kernels and other memory quantifiers, etc. None of these requires any kind of underlying model, all of these directly provide physical intuition, in contrast to the proposed new approach.

The motivation and declared superiority (if any at all) of the proposed black-box approach must be quantified by a fair benchmark comparison (which may be difficult), or the statements claiming superiority dropped. Moreover, any kind of such statements must also be formulated precisely and factually. No comparison analysis is performed whatsoever. It is not clear if the above intuitive “canonical” measures for quantifying the properties of trajectories (and their differences) are truly outperformed by the present approach to a degree that would outweigh the fact that the black-box treatment offers little if any physical insight.

2) The existing literature on quantifying properties of individual particle traces is literally not existent. On the one hand this is nominally unacceptable. On the other hand it may explain the authors' belief that the task requires to identify some underlying model in the sense that they are perhaps simply not familiar with the literature well enough to recognize that this is not true.

3) Consider the following conclusions stated in the manuscript:

“This indicates that the addition of KCl to the medium can affect the physical properties of many if not all alpha-synuclein molecules in a similar manner, irrespective of their subcellular location.”

“This demonstrates that alpha-synuclein is highly mobile in living cells …”.

“… the representative αlpha-synuclein trajectories exhibit a greater mobility in the depolarised state. This is likely the result of a weaker binding of alpha-synuclein at synapses, as reflected in the overall reduction of alpha-synuclein molecules during KCl application ...”

Any of the above mentioned canonical analyses would have directly reveal this information. In other words, the authors seem to have found a (very indirect) black-box substitute to infer physics. I am no biologist but these conclusions also do not seem to provide any new biological insight.

4) The title and discussion claims “subtle differences in alpha-synuclein dynamics”. I am no biologist (perhaps this may be required) but I really struggled to identify why the observed differences are “subtle”. Based on the title I expected that all canonically used methods employed in studies of particle transport (incl. the more “advanced” approaches) would fail to the detect differences, and this would call for the proposed approach. I would agree that such differences were subtle. But since no true comparison is presented the word “subtle” does not seem to be justified.

Summarizing remarks:

The results may certainly be interesting for a specialized community interested in technical aspects of the analysis of particle-tracking data. However, the analysis does not seem to be really required (at least with the motivation given in the manuscript). The way the results are presented one may be led to thinking that this is the kind of analysis to be performed if one aims to identify differences between measured trajectories but one does *not* want to gain the physical insight that comes “for free” in the application of traditional physics-based analyses. I may be mistaken, but the way the results are presented simply leads to such a conclusion.

As a result, based on this version of the manuscript I tend to recommend against publication and to instead seek a more specialized journal.

**Have the authors made all data and (if applicable) computational code underlying the findings in their manuscript fully available?**

Reviewer #1: **No: **The code and the data can't be accessed

Reviewer #2: Yes

PLOS authors have the option to publish the peer review history of their article (what does this mean?). If published, this will include your full peer review and any attached files.

Reviewer #1: No

Reviewer #2: No
---

## [Decision Letter · Decision Letter 1]

10 Nov 2022

Dear Mr. Verdier,

Thank you very much for submitting your manuscript "Simulation-based inference for non-parametric statistical comparison of biomolecule dynamics" for consideration at PLOS Computational Biology. As with all papers reviewed by the journal, your manuscript was reviewed by members of the editorial board and by the original reviewers. The reviewers appreciated the improvements in the revised manuscript, but also expressed a few remaining concerns. Based on the reviews, we are likely to accept this manuscript for publication, providing that you modify the manuscript according to the review recommendations

Sincerely,

Bert L. de Groot

Academic Editor

PLOS Computational Biology

Arne Elofsson

Section Editor

PLOS Computational Biology

Reviewer's Responses to Questions

**Comments to the Authors:**

Reviewer #1: The revision has largely improved the manuscript but has produced a major shift in the focus of the article, which now reads as an example of a general recipe about how to handle relatively large datasets involving heterogeneous entities (in this case, sets of short single-molecule trajectories) by extracting meaningful features (using GNN, others ML approaches, or even analytical indicators), statistically testing them for (typically small) differences, and highlighting regions of the latent spaces that might produce these differences.

Due to the recent development of high-throughput methods for live-cell imaging and the inherent biological variability of these experiments, I believe this work is relevant to the quantitative development of the field.

Still, there are a few important points that, in might opinion, need to be further discussed and clarified. Since the authors are now focusing on the methodology and on its generality, I believe that it would be beneficial to provide additional insights on the statistical test and the simulation-based inference steps.

Major points:

1) To be fair, one should compare the results of the MMD test to those obtained through another nonparametric multivariate (or univariate, respectively) 2-sample test, whereas the Hotelling T-squared (or t-test, respectively) is parametric and requires normality and homoskedasticity.

2) The experiments described in the article involve the comparison of more than two conditions. In this case, a 2-sample test can be used as a post-hoc pairwise analysis (with the proper correction for multiple comparisons) after carrying out a (significant) nonparametric omnibus test, whereas the authors apply directly the MMD test pairwise. Please discuss this point and propose a procedure for these cases.

3) I can agree with the authors that different architectures showing similar performance in an inference or classification task learn an equivalent latent representation. However, I believe that the relevant question here is how the choice of the learning bias influences the latent representation. For example, I would expect the method to behave differently if one would use different (or additional, or less) target features for training with respect to model probability and alpha. Or if one would replace the supervised GNN with an architecture with no additional target features besides trajectory coordinates and time. What would the machine learn in these cases? How would the latent representation change? Just to give an example, one might think to obtain the latent representation by using autoencoders (as in https://iopscience.iop.org/article/10.1088/1751-8121/ac3786) or even a generative model (e.g., variational autoencoders).

4) I wonder how much the choice of the dataset used for the training contributes to implicitly defining the latent features that are extracted. I guess it would be interesting to investigate the influence of the parameter space used in the training dataset. I believe that some comparison (e.g., using different training sets and the same test set) might help to partly clarify this point.

5) Are the learned features correlated? The results shown in fig S4B (Gratin2 vs Gratin16) seem to suggest so. Please discuss the possible implications for the statistical test.

6) Are the learned features correlated with the analytical indicators (before UMAP reduction)? This might provide hints about the interpretability of the features.

Minor points:

1) In the reply, The authors stated “We also added a reference to the method of Muñoz-Gil et al. 2021, which, like ours, relies on learnt representations of individual trajectories” but I could not find it in the references of the revised manuscript.

2) There is a missing reference on line 816

3) I find it confusing to use \\alpha both for the anomalous diffusion exponent and the significance level of the statistical test

Reviewer #2: The authors did a good job in revising their manuscript, but a few points in their response are rather obscure. In point 1 they claim that "Very often, observed biomolecules do not all exhibit the same type of dynamics ... " which I must interpret as a statistical ensemble of paths (and a probability measure on said space of paths) does not exists. Few sentences later they claim that their method can "identify the most closely corresponding random walk model" which directly (and irrefutably) contradicts the previous statement. I know that the probabilistic "hygiene" is not the main priority, but the authors should try being a bit more coherent in their claims.

Moreover, stationary is not an assumption that one must make, one can test for that (directly, and rather straightforwardly). Scientists have been (quantitatively) describing observations of glassy behavior long before the advent of machine learning.

I do agree that the "dealing with under-sampled data" may be a strength of the method and this could potentially be made even more visible.

Something I did not mention in my original report (but the authors may consider) is to classify trajectories of the "unequal twin" processes using a small number of trajectories (https://journals.aps.org/prl/abstract/10.1103/PhysRevLett.107.260601). This may be a challenge but would (potentially) also objectively demonstrate the power of the new method. But this is optional.

**Have the authors made all data and (if applicable) computational code underlying the findings in their manuscript fully available?**

Reviewer #1: Yes

Reviewer #2: Yes

PLOS authors have the option to publish the peer review history of their article (what does this mean?). If published, this will include your full peer review and any attached files.

Reviewer #1: No

Reviewer #2: No

Figure Files:

Data Requirements:

Reproducibility:

References:

---

## [Editor Report · Decision Letter 2]

16 Jan 2023

Dear Mr. Verdier,

We are pleased to inform you that your manuscript 'Simulation-based inference for non-parametric statistical comparison of biomolecule dynamics' has been provisionally accepted for publication in PLOS Computational Biology.

Best regards,

Bert L. de Groot

Academic Editor

PLOS Computational Biology

Arne Elofsson

Section Editor

PLOS Computational Biology

---

## [Editor Report · Acceptance letter]

26 Jan 2023

PCOMPBIOL-D-22-00557R2 

Simulation-based inference for non-parametric statistical comparison of biomolecule dynamics

Dear Dr Verdier,

I am pleased to inform you that your manuscript has been formally accepted for publication in PLOS Computational Biology. Your manuscript is now with our production department and you will be notified of the publication date in due course.

With kind regards,

Bernadett Koltai
